# DIRECT REWARD OPTIMIZATION: A POINT-WISE ALIGNMENT APPROACH

## ABSTRACT

Direct Alignment Algorithms (DAAs) are widely used for aligning Large Language Models (LLMs) with human preferences. The current DAAs mostly use pairwise optimization objectives based on variants of Direct Preference Optimization (DPO). However, these methods only focus on the pairwise differences of the samples and cannot prevent the optimization from reducing the probabilities of preferred responses. To tackle the problem, in this paper, we propose Direct Reward Optimization (DRO), an algorithm that uses an explicit reward model to optimize the policy by setting an exact probability target for each response. DRO decouples the target reward differentials and bias in aligning objectives and utilizes the relationships not only within but also among the response pairs. Extensive experiments show that DRO outperforms the existing methods while providing control over the policy response probability.

## 1 INTRODUCTION

Large Language Model (LLM) alignment aims to enhance the model's ability to align with human values and preferences. A typical LLM alignment approach is Reinforcement Learning from Human Feedback (RLHF) Ouyang et al. (2022). RLHF relies on annotated preference data (i.e. positive and negative response pairs) to model human preferences through the Bradley-Terry (BT) model Bradley & Terry (1952). Although RLHF has achieved a state-of-the-art performance, its pipeline is complex, involving the training of multiple LLMs and sampling processes within the training loop. Thus, simpler alignment methods, known as Direct Alignment Algorithms (DAAs), are becoming the mainstream approach Gupta et al. (2025).

DAAs primarily incorporate the Direct Preference Optimization (DPO) Rafailov et al. (2024) and various adaptations of it. DPO reparameterizes the reward function within the RLHF framework, suggesting that the optimizing policy can act as an implicit reward function. It increases the generation probability gap of preference pairs by leveraging the BT model.

Although DPO shares the same optimization objective and shows comparable performance with RLHF, the BT training paradigm (i.e., optimizing the probability gap) brings about several problems Meng et al. (2024); Sharifnassab et al. (2024); Lin et al. (2024). First, it optimizes the implicit reward model differently from the explicit reward model, leading to a side effect on the generation distribution of the policy model (i.e., likelihood displacement). When the lower strength of KL constraint is used (i.e., a small $\beta$), DPO simultaneously reduces the probabilities of preferred responses and dispreferred responses, while increasing their gap Meng et al. (2024); Hong et al. (2024). Too low probabilities of preferred responses can result in the LLM not being inclined to generate similar responses, leading to a negative impact on policy Gupta et al. (2025). Nonetheless, this does not happen on the BT optimization of explicit reward model. Current approaches tend to solve the problem by adding different weights to the preferred and dispreferred responses in the training objective Gupta et al. (2025); Hong et al. (2024). However, this breaks the consistency between the objectives of DPO and RLHF. Moreover, the added hyperparameters require additional costs to locate the proper values in specific tasks.

Second, the structure of the implicit reward model restricts the performance of the BT training paradigm. Recent research Lin et al. (2024) points out that the implicit reward model shows limited generalization capability (compared to explicit reward model training under the BT model in RLHF). This indicates that the BT training paradigm may not be suitable for the implicit reward

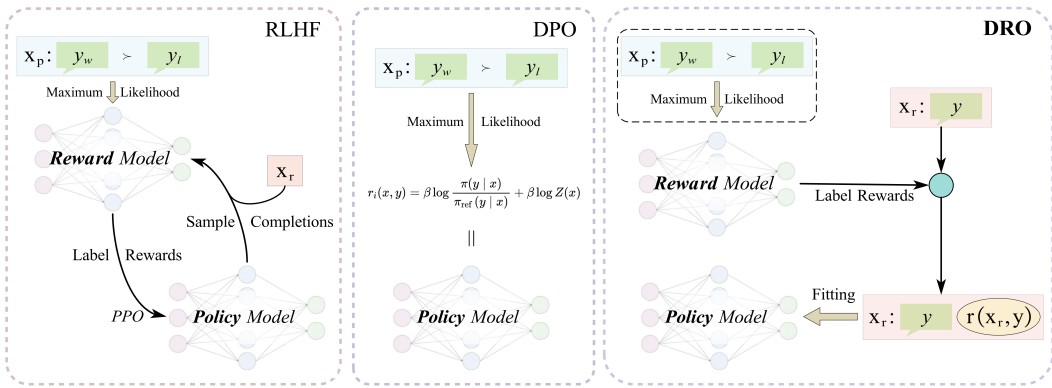

Figure 1: Differences of the proposed DRO from RLHF and DPO. RLHF trains a reward model using the BT model and applies PPO to optimize the policy model online. DPO uses the BT model to optimize the policy model offline. DRO uses a reward model (which may be trained by the BT model) to annotate the responses and offline distills the reward to the policy model.

model. Various methods Adler et al. (2024); Fisch et al. (2024) attempt to alleviate this problem by introducing an explicit reward model to the DPO and show an improved performance. However, these methods do not consider the drop in the probabilities of preferred responses as discussed above. Moreover, they only take into account the reward differences between the responses within a pair, yet ignoring the relationship among sample pairs indicated by the given reward.

Third, as LLMs' capabilities continue to get strengthened, the BT training paradigm gradually shows inferiority in learning complex reward distributions. The structure of the reward model, which can be optimized by BT training paradigm (e.g., the sequence classification-based reward model, the implicit reward model), contains no Chain-of-Thought process Zhang et al. (2024). Consequently their reward modeling capabilities are inadequate for complex tasks (i.e., reasoning tasks). Thus, the adoption of GenRM Mahan et al. (2024); Mahan et al.; Seed et al. (2025)(generative reward model) in reinforcement learning has become more frequently considered. However, for DAAs where the BT model serves as a theoretical basis, there is a lack of work that directly discusses this optimization dilemma.

In this paper, we aim to address these limitations by answering the question: **Can DAA optimize the policy by directly guiding the exact target of generation probability (instead of adopting BT training paradigm)?** We observe that current DAAs adopt pairwise optimization structure to eliminate the normalization terms in deriving the RLHF objective for each sample (detailed in Section 2). In this paper, we find through a theoretical analysis that these terms can be derived from an invariant value and the optimal policy. By regarding this value as a hyperparameter, we propose a Direct Reward Optimization (DRO) algorithm, which uses an explicit reward model to offline optimize the policy by point-wise distilling the reward to an implicit reward model.

Compared to current DAAs, DRO dislodges the pair-wise BT training paradigm and adopts a point-to-point optimization form. We utilize an explicit reward model (reveal the human preference estimation) to set an exact target probability for each response, and use point-to-point MSE loss for optimization. There are remarkable advantages shown by DRO. First, DRO naturally avoids the problem of decreased probabilities for preferred responses. In practice, our method decouples the target reward differentials and offsets of DAA, and provides controllability to the training target of the policy model's probability distribution. Thus, our approach offers practitioners more flexibility in adjusting optimization targets. On the other hand, our DRO better utilizes the explicit reward model (compared to previous approaches), referring not only to the relationship between the responses with the same prompt but also to the relationship among the responses with different prompts. Meanwhile, the simplicity of DPO is preserved. Furthermore, our DRO allows the adoption of reward models in various forms (e.g., GenRM or rule-based reward), ensuring that DRO is not affected by the training of reward model.

In summary, our main contribution lies in the Direct Reward Optimization (DRO), a novel pair-wise optimization-free direct alignment algorithm with an explicit reward model. The model decouples the target reward differentials and bias, and fully utilizes the reward information. Our experiments show that DRO largely outperforms the existing methods with respect to a range of base LLMs, including Llama3-8B Dubey et al. (2024), Qwen2.5-7B Yang et al. (2024), and EuroLLM-9B Martins et al. (2024).

## 2 PRELIMINARIES

Given a large language model parameterized by $\theta$, donated as $\pi_\theta$. The current alignment algorithms aim to optimize $\pi_\theta$ by learning from annotated preference pairs.

**RLHF:** RLHF Bai et al. (2022) fits a reward model to pairwise samples of human preferences and then uses Proximal Policy Optimization (PPO) to optimize a language model policy to produce responses that are assigned a higher reward without drifting excessively far from the original model. Consider an annotated dataset of pairwise samples $\mathcal{D}_p = \left\{ x_i, y_w^i, y_l^i \right\}_{i=1}^N$, where $x_i$ denotes the $i^{th}$ prompt, $y_w^i$ and $y_l^i$ respectively represent the preferred and dis-preferred responses to $x_i$. RLHF begins with modeling the probability of preferring $y_w^i$ to $y_l^i$ using the Bradley-Terry (BT) model Bradley & Terry (1952), which appoints the following probabilistic form:

$$p \left( y_w^i \succ y_l^i \mid x \right) = \sigma \left( r \left( x_i, y_w^i \right) - r \left( x, y_l^i \right) \right) \tag{1}$$

where $\sigma$ represents the logistic function and $r(x_i, y_i)$ corresponds to a reward function $r_\phi$ (that is, the LLM classifier) that gives the estimation of $y_i$ with respect to $x_i$ according to human preference. Using maximum likelihood estimation to estimate the parameters of this function, we can optimize the classifier by the negative log-likelihood loss as below:

$$\mathcal{L}_R \left( r_\phi, \mathcal{D} \right) = -\mathbb{E}_\mathcal{D} \left[ \log \left( \sigma \left( r_\phi \left( x, y_w \right) - r_\phi \left( x, y_l \right) \right) \right) \right] \tag{2}$$

The target model $\pi_\theta$ can then be trained by the feedback of the learned reward function. In general, we formulate the following optimization target for this learning process.

$$\max_{\pi_\theta} \mathbb{E} \left[ r_\phi(x, y) \right] - \beta \mathbb{D}_{\text{KL}} \left[ \pi_\theta(y \mid x) \| \pi_{\text{ref}}(y \mid x) \right] \tag{3}$$

where $\beta$ is a parameter that controls the deviation of the target model $\pi_\theta$ from the status when training starts.

**DPO:** DPO Rafailov et al. (2024) shows the possibility of keeping the same optimization target as the RLHF's without explicitly training a reward function and the implementation of RL. The loss function of DPO is presented below:

$$\mathcal{L}_{\text{DPO}} \left( \pi_\theta; \pi_{\text{ref}} \right) = -\mathbb{E}_{(x, y_w, y_l) \sim D} \log \sigma \left( \beta \log \frac{\pi_\theta \left( y_w \mid x \right)}{\pi_{\text{ref}} \left( y_w \mid x \right)} - \beta \log \frac{\pi_\theta \left( y_l \mid x \right)}{\pi_{\text{ref}} \left( y_l \mid x \right)} \right) \tag{4}$$

Notably, this optimization objective is based on a theoretical optimal $\pi_\theta$ beyond $r_U(x, y)$, which enables its equivalence with Eq.3.

## 3 DIRECT REWARD OPTIMIZATION

This section presents the proposed Direct Reward Optimization (DRO) algorithm. Aiming to guide the exact probability of responses for the policy LLM, we derive our training objectives from RLHF, referring to the previous work Rafailov et al. (2024) and introduce a reward model. By referring to the normalization term as a hyperparameter, DRO distills the reward of an explicit model into the implicit reward of policy LLM.

### 3.1 REWARD MODEL

DRO uses the reward model to distill the rewards of an offline dataset into the policy LLM to guide the LLM to become the optimal policy under the objective Eq. 3. This ensures our DRO relies on a reward model with better generalization capability compared to the DAA's training implicit reward model through the BT training paradigm. Furthermore, our point-wise optimization utilizes the reward relation between responses with different prompts rather than pair-wise DAA.

Notably, DRO is not restricted to one specific reward model training method. That means that DRO accepts GenRM or a rule-based reward model for optimization. However, in our experiments, for reward model training, we follow the RLHF utilizing a Bradley-Terry model to model the preference of a pair-wise dataset Rafailov et al. (2024) in order to focus on comparing the advantages of DRO in alignment optimization rather than reward models. Specifically, we use Eq. 2 to train a neural reward model that uses a classifier to process the hidden state of the last token given by a pretrained LLM.

### 3.2 DIRECT REWARD OPTIMIZATION

Starting from the RLHF objective, we follow the previous work Rafailov et al. (2024) and construct the implicit reward function under the optimal solution $\hat{\pi}$ to Eq. 3 as follows:

$$r(x, y) = \beta \log \frac{\hat{\pi}(y \mid x)}{\pi_{\text{ref}}(y \mid x)} + \beta \log Z(x) \tag{5}$$

where $Z(x) = \sum_y \pi_{\text{ref}}(y \mid x) \exp\left(\frac{1}{\beta} r(x, y)\right)$ represents the normalization term. Due to space limitations, we present a detailed derivation process in the Appendix A.1.

The normalisation term $Z(x)$ changes with prompts $x$, resulting in the result that the implicit reward target needs exact $\mathbf{Z} = \{Z(x_1), Z(x_2), ..., Z(x_n)\}$. Considering that the given rewards are the same for different splits of x and y in the sentence "xy", we can deriving a relationship between $Z(x, y)$ (which means the $Z("xy")$ referring $r(x, y)$) and $Z(x)$ as below (detailed derivation process is presented in the Appendix C):

$$\frac{Z(x, y)}{Z(x)} = \frac{\hat{\pi}(y \mid x)}{\pi_{ref}(y \mid x)} \tag{6}$$

Through this relationship, we can assume an imaginary overall prefix $t_0$ that fits every prompt $x_i$. Thus the normalization term $Z_0 = Z(t_0)$ whose defination is $Z_0 = \sum_y \pi_{\text{ref}}(y \mid t_0) \exp\left(\frac{1}{\beta} r(t_0, y)\right)$. This indicates that the relationships among $\mathbf{Z}$ are related to the $\hat{\pi}$ and $\pi_{ref}$. Once obtaining the value of $Z(x_i)$, our DRO optimize the policy utilizing the MSE Loss:

$$\mathcal{L}_{DRO}(\pi_\theta, r, \mathbf{Z}; \mathcal{D}) = \mathbb{E}_{(x,y) \sim \mathcal{D}}\left[\left(r(x, y) - \beta \log \frac{\pi(y \mid x)}{\pi_{\text{ref}}(y \mid x)} - \beta \log Z(x)\right)^2\right] \tag{7}$$

### 3.3 OPTIMIZATION

DRO distils the explicit reward to improve the LLM policy. Referring to the work of Adler et al. (2024), we adopt the phase of including more than one response per prompt for training to ensure better preference supervision. Notably, while the assumption of $Z_0$ requires an overall prefix $t_0$ which every prompt $x_i$ has, DRO theoretically restricts the prompts to have the same "start token". It is easy to meet this condition since almost every LLM template stipulates the first token (e.g., "$\langle|\text{im\_start}|\rangle$" or "User").

**Theorem 3.1.** *Suppose a reward model $r(x, y)$ and a dataset $\mathcal{D} = \{x_i, y_i\}_{i=1}^N$, infinite various $\bar{r}_r$ can be constructed ensuring: 1. $r(x_i, y_i) = \bar{r}_r(x_i, y_i)$ for $x_i, y_i \in \mathcal{D}$. 2. For all $x_1, y_1, x_2, y_2$ in the language space, $[r(x_1, y_1) - r(x_2, y_2)][\bar{r}_r(x_1, y_1) - \bar{r}_r(x_2, y_2)] > 0$*

---

**Algorithm 1:** Direct Reward Optimization

---

**Input:** SFT model $\pi_\theta$, Reward model $r$, Training Data $\mathcal{D}$, Norm Value $Z_0$ with $t_0$, Training
       Epochs T, Learning Rate $\eta$

**Output:** Optimized Policy $\hat{\pi}_\theta$

1   $\pi_{ref} \longleftarrow \pi_\theta$ ;

2   **foreach** *Epoch t=1, 2, ..., T* **do**

3      Get a batch of samples $\mathcal{D}_\sqcup \subset \mathcal{D}$ ;

4      **foreach** $\left(x_i, y_i^1, y_i^2, ...\right) \in \mathcal{D}_\mathcal{B}$ **do**

5         $\mathcal{L}_\mathcal{T} \longleftarrow 0$ ;

6         $Z_i \longleftarrow \frac{\pi_\theta(x_i - t_0 | t_0)}{\pi_{ref}(x_i - t_0 | t_0)} Z_0$ ;

7         Detach $Z_i$ ;

8         **foreach** $y_i^j$ **do**

9            $r_j \longleftarrow \beta \log \frac{\pi_\theta(y_i^j | x_T)}{\pi_{\text{ref}}(y_i^j | x_i)} + \beta \log Z_i$ ;

10          $\mathcal{L}_\mathcal{T} \longleftarrow \mathcal{L}_\mathcal{T} + \left(r(x_i, y_i^j) - r_j\right)^2$ ;

11         $\pi_\theta \longleftarrow \pi_\theta - \eta \nabla \left(\frac{\mathcal{L}_\mathcal{T}}{|\text{the number of } y \text{ for } x_i|}\right)$ ;

12   $\hat{\pi}_\theta \longleftarrow \pi_\theta$;

---

While the actual value of $Z_0$ cannot be calculated by its definition since the possible $y$ is infinite, in DRO, we regard it as a hyperparameter. As our derivation in App. D proving Thm. 3.1, there're different reward models having different $Z_0$ act the same in the optimization. The value of $Z_0$ acts as a controller of reward value bias in DRO, for which we will present an analysis in Sec. 4.5. We approximate $\hat{\pi}$ to $\pi_\theta$ in optimization, ensuring the consistency of the optimal solution of DRO. The experiments further confirm that this approximation does not compromise the convergence.

We use a pseudocode presented as Algorithm 1 to show the DRO optimization. DRO aims to optimize the implicit reward of the policy and treats the normalization term $Z_i$ as a constant. After obtaining the target of $\log \hat{\pi}(y \mid x)$, DRO utilizes an MSE loss for training, referring to previous work Fisch et al. (2024).

### 3.4 THE INTERPRETATION OF DRO

Our DRO utilizes Eq. 6 to generate an approximate normalized term to Eq. 5 and uses the MSE loss for optimization. While combining $Z(t_0)$ to Eq. 5 using Eq. 6, we can result to the below equation:

$$r(x,y) = \beta \log \frac{\hat{\pi}(t_0 \mid (x,y) - t_0)}{\pi_{\text{ref}}(t_0 \mid (x,y) - t_0)} + \beta \log Z(t_0) \tag{8}$$

Which is the Eq. 5 in a certain situation. In particular, in Algo. 1, $Z_i$ does not contribute to the gradient since the generation probabilities of the prompts are within our optimization scope, which makes DRO optimization different from the direct utilization of Eq. 8.

As Eq. 8 shares the same optimal policy with DRO, we can infer from it that $\beta$ presents the level of reward differences of our optimization target. The smaller $\beta$ is, the greater the gap among our reward target, which is the same in the work of Rafailov et al. (2024). $Z_0$ in DRO presents an "offset" to the rewards. While $Z_0$ grows down, all the reward targets move upwards. This ensures that DRO controls the generation probabilities from simultaneous decreases.

## 4 EXPERIMENTS

We experiment with our DRO based on the below pretrained LLMs: Llama3-8B Dubey et al. (2024), Qwen2.5-7B Bai et al. (2023), and EuroLLM-9B Martins et al. (2024). In this section, our aim is to present the advantages of our DRO versus other direct alignment baselines. We start from the base

| Model Setting | Small | | Large | |
|---|---|---|---|---|
| | Loss | Acc | Loss | Acc |
| RM-Base | 0.0621 | 0.975 | 0.0539 | 0.982 |
| RM-SFT | 0.0463 | 0.979 | 0.035 | 0.988 |
| DPO-Implicit | 0.2039 | 0.9521 | 0.2463 | 0.9660 |

Table 1: The reward model training results.

models and finetune them to gain the instruction-following capability. Reward models are trained on a pairwise preference dataset. Then, we use the reward models to annotate the rewards of this preference dataset, and DRO is used to optimize the finetuned LLMs. Notably, we keep sampling two responses each prompt in order to keep the scale of training data the same as DRO and all baselines. Notably, we put our Implementation Details in the App. B

### 4.1 DATASETS AND EVALUATIONS

We follow the typical training pipeline of Zephyr Tunstall et al. (2023) and SimPO Meng et al. (2024) to select datasets. For the supervised finetuning phase, we apply the UltraChat-200k dataset Ding et al. (2023) to train our base models. Notably, we optimize the base models utilizing the multi-turn dialogue templates of their chat derivatives. For reward model training and alignment optimization, we apply the UltraFeedback dataset Cui et al. (2023). This approach provides a high level of reproducibility. We present an introduction of these datasets in the Appendix E.

For evaluation benchmarks, we apply the widely used benchmarks for general instruction-following capability: Alpaca-Eval2 Dubois et al. (2024) and MT-Bench Zheng et al. (2024). These benchmarks evaluate the LLM's versatile conversational capabilities utilizing different queries. Alpaca-Eval2 constructs its 805 queries from 5 datasets, and MT-Bench contains 80 queries sampled from 8 different categories. Both benchmarks rely on an LLM-as-judge evaluating methods. Notably, we use GPT-4 Achiam et al. (2023) as the annotator for them. For Alpaca-Eval2, we present the results of win rate (WR) and length-controlled win rate (LC), which reflects the evaluation results eliminating the effect of model verbosity over a base response, which is sampled from GPT-4 Turbo Achiam et al. (2023). For MT-Bench, we report the average overall score calculated based on the judgment of GPT-4.

### 4.2 BASELINES

We compare our DRO with different direct alignment algorithm baselines. Except for the widely used and introduced **DPO** and **RLHF**, **IPO** Azar et al. (2024) constructs a general preference learning structure objective, deriving from which DPO is a special case, bypasses the BT modelization assumption for preferences, and utilizes an MSE loss. **ORPO** Hong et al. (2024) drop the reference model in DPO and introduce an odd ratio to directly optimize the probabilities of the policy model while jointly training with an objective of preferred response maximum likelihood loss. **SimPO** Meng et al. (2024) uses the average log probability of a sequence as the implicit reward and introduces a target reward margin in the DPO objective. **Robust Preference Optimization (RPO)** Fisch et al. (2024) introduces an explicit reward model to distill the reward gaps to the policy model. Notably, we use the same reward model to address the reward gaps as our DRO uses. We only use one reward model in RPO to ensure the fairness of our DRO and RPO. **Cal-DPO** Xiao et al. (2024) introduces IPO-like calibration loss to DPO loss to solve likelihood displacement. Notably, except for IPO, all the above methods do not share the same optimal solution consistency as DPO and DRO with RLHF. We report the hyperparameter search area of each baseline in Appendix F.

### 4.3 REWARD MODEL

Our DRO doesn't specify the approach of the reward model used to give the reward. Here we present a demonstrative reward model training process. We utilize the Bradley-Terry model to train an explicit reward model that gives a reward score through a randomly initialized classifier on the hidden state of the last token of a pretrained model's output. To compare the performances of explicit reward models initialized with the base model the SFT model and the implicit reward model indicated in Eq. 5, we utilize all the preference pairs in UltraFeedback (regarded as "large" setting)

| Methods | Llama3-8B | | | Qwen2.5-7B | | | EuroLLM-9B | | |
|---|---|---|---|---|---|---|---|---|---|
| | AlpacaEval 2 | | MT-Bench | AlpacaEval 2 | | MT-Bench | AlpacaEval 2 | | MT-Bench |
| | WR(%) | LC(%) | | WR(%) | LC(%) | | WR(%) | LC(%) | |
| SFT | 3.35 | 5.82 | 5.0 | 5.41 | 10.69 | 5.7 | 4.11 | 7.81 | 5.3 |
| DPO | 18.32 | 17.63 | 6.5 | 18.12 | 23.16 | **6.8** | 12.52 | 16.02 | 6.0 |
| RLHF | 16.08 | 15.97 | 6.3 | 16.78 | 19.54 | 6.5 | 12.71 | 15.85 | 6.0 |
| IPO | 14.92 | 15.24 | 6.1 | 13.25 | 14.47 | 6.4 | 11.38 | 11.98 | 5.8 |
| ORPO | 11.97 | 13.535 | 5.7 | 9.10 | 12.72 | 6.2 | 9.29 | 12.26 | 5.8 |
| SimPO | 18.42 | 19.97 | **6.6** | 17.32 | 23.28 | 6.7 | **14.92** | 16.53 | **6.2** |
| RPO | 18.52 | 19.24 | **6.6** | 17.74 | 22.14 | 6.6 | 14.24 | 14.59 | 6.1 |
| Cal-DPO | 18.51 | 17.02 | 6.5 | 18.27 | 22.09 | 6.7 | 13.25 | 16.72 | 6.1 |
| DRO | **19.51** | **21.94** | **6.6** | **20.82** | **26.04** | **6.8** | 14.11 | **17.64** | **6.2** |

Table 2: Main Results on UltraFeedback Dataset. The best performance is in bold.

| Methods | MMLU | GSM8K | ARC-Easy | ARC-Hard | MathQA | SocialQA | Avg. |
|---|---|---|---|---|---|---|---|
| SFT | 63.81 | 25.84 | 52.82 | 48.29 | 26.73 | 50.25 | 44.62 |
| DPO | 64.88 | 24.84 | 49.37 | 39.25 | **28.88** | 37.45 | 41.45 |
| RLHF | 63.82 | 25.77 | 56.72 | 50.17 | 26.81 | 53.48 | 46.92 |
| IPO | 63.25 | 28.96 | 60.29 | 45.30 | 27.03 | 40.78 | 44.27 |
| ORPO | 65.02 | 26.24 | 63.95 | 49.82 | 24.14 | 53.69 | 47.14 |
| SimPO | 63.47 | 25.02 | 44.57 | 36.6 | 25.42 | 36.83 | 38.65 |
| RPO | 64.32 | 26.37 | 58.33 | 51.50 | 26.29 | 52.20 | 46.50 |
| Cal-DPO | 64.68 | 27.25 | 58.39 | 51.27 | 27.01 | 52.85 | 46.91 |
| DRO | **65.25** | **31.72** | **69.49** | **55.38** | 27.19 | **54.95** | **50.66** |

Table 3: Overall result in the Downstream Tasks for Models trained on UltraFeedback. The best performance is in bold.

or 10000 pairs randomly sampled from it (regarded as "small" setting) either to train the reward models based on Llama3. Taking the loss of training ends and the metrics of reward accuracy (i.e., the accuracy of the reward model gives a larger reward to preferred responses than dispreferred ones) on the test set of UltraFeedback, we present the results in Tab. 1.

We can observe that the explicit reward model initialized by the SFT model performs best among the three. The either explicit model shows an apparent advantage to the implicit model. This indicates the benefits of using an explicit reward model for alignment as our DRO. Following the results, we train the reward model of Qwen2.5 and EuroLLM using their SFT model instead of directly using the base model. Specifically, we use the same reward model for DRO and RLHF.

## 4.4 MAIN RESULTS

The main results of our experiments are presented in Tab. 2. Remarkably, while all the direct alignment baselines optimize the SFT model to a better conversational capability, referring to the benchmarks, DRO outperforms all the baselines in all settings except SimPO on EuroLLM-9B on the Alpaca-Eval 2 win-rate metric. This illustrates the advantages of DRO over current alignment methods. Notably, DRO achieves an 82.83% increase over the SFT model and a 5.04% increase over RPO, which performs best among the baselines in the Alpaca-Eval 2 win rate metric based on Llama3-8B, and this advantage comes to 73.47% and 12.31% on the length-controlled win rate. For Qwen2.5-7B, DRO gains 14.79% and 14.98% advantages compared to the best baseline on win rate and length-controlled win rate of Alpaca-Eval 2. For EuroLLM-9B, DRO gains a 6.29% advantage on the length-controlled win rate.

A cursory examination reveals that our DRO has an obvious outperformance over all the direct alignment baselines across all tasks. Such a pattern underscores the effectiveness of DRO in improving LLM's ability in preference learning. DRO not only introduces an explicit reward model that has a better generalization capability to the alignment training but also provides a more stable training target using point-wise loss and prevents the continual decrease of preferred response probabilities.

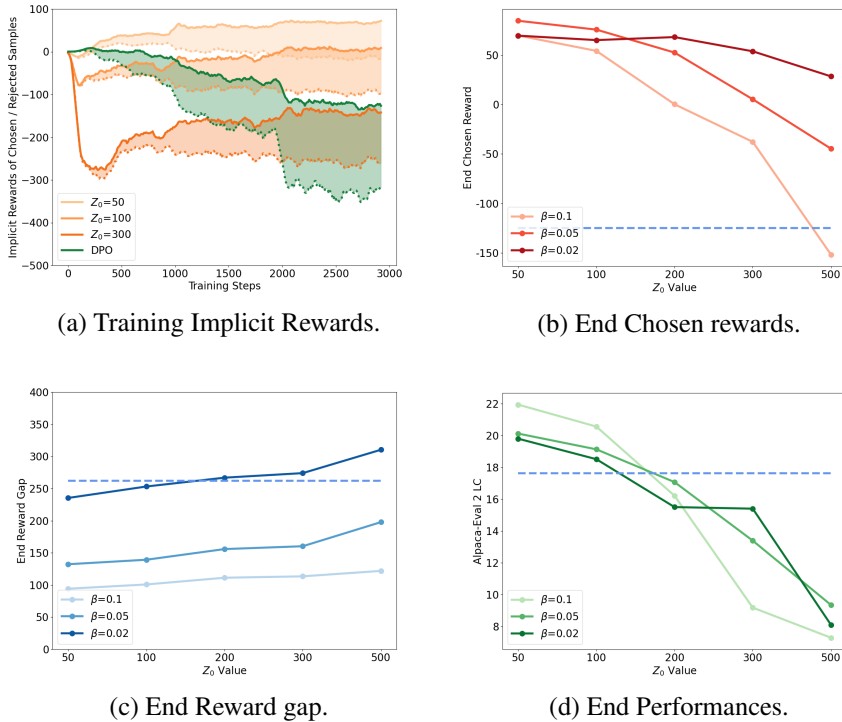

(a) Training Implicit Rewards.

(b) End Chosen rewards.

(c) End Reward gap.

(d) End Performances.

Figure 2: Analysis of DRO training process. The analysis experiments are conducted on Llama3-8B under different hyperparameters. The blue dashed line represents the performance of DPO. The dotted line in (a) indicates the reject samples' reward and the solid line indicates the chosen samples' reward.

## 4.5 ANALYSIS

We present here a detailed analysis of our DRO controls and the alignment process of the policy. As shown in Fig. 2, we conclude:

- DRO utilizes point-wise loss to optimize the policy model. It sets a target for the chosen reward of the policy model; thus, we can observe from Fig. 2(a) that the rewards of both chosen and rejected rewards are symmetrically separated from each other while keeping a clear, stable mean value. This mean value is the $Z_0$ value set to be stable in the training process. While $Z_0$ grows larger, the value drops.

- From another perspective, the effect of $Z_0$ and $\beta$ in DRO is more clearer in Fig. 2(b). While $Z_0$ grows larger, the chosen reward of the training end decreases. While $\beta$ grows smaller, this decreasing trend becomes slower. It can be inferred that when $Z_0$ is sufficiently larger, the chosen reward can be smaller than utilizing DPO.

- As for the gap between chosen rewards and rejected rewards in the training ends, $\beta$ can have a significant effect. While $\beta$ drops, this gap grows rapidly. One of our DRO's main effectiveness is decoupling the reward gap and the mean value of the alignment target. It can be seen in Fig. 2(c) that $Z_0$ has little effect on the reward gap.

- From Fig. 2(d) we can observe that the performance of the alignment algorithm is affected by the combination of other factors. Neither reward gap nor the chosen reward can reflect the final performance independently.

## 4.6 DOWNSTREAM TASKS EVALUATION

To examine how exactly the models perform in different fields, we evaluate all the models reported in Tab. 2 which is based on Llama3-8B to various downstream tasks. Specifically, we include the

MMLU Hendrycks et al. (2020), GSM8K Cobbe et al. (2021), ARC-Easy and Challenge Clark et al. (2018), MathQA Amini et al. (2019), and SocialQA Sap et al. (2019). As reported in Meng et al. (2024), several direct alignment algorithms may drop the model's performance in reasoning tasks. Thus, we mainly choose the reasoning tasks in our evaluation and the widely used MMLU. Notably, except for MMLU, all the tasks are evaluated through the CoT Pass@1 zero-shot setting. We set the sampling temperature to 0.0 to adopt the greedy sampling method.

The results are presented in Tab. 3. We can observe that DRO performs better than all the baselines. While alignment methods as DPO and SimPO obviously drop the model's reasoning capabilities, DRO does not decrease the ability of the SFT model and instead improves the reasoning ability of the model through alignment. We infer that some baselines dropping the model's reasoning capability may be caused by the significant decrease of preferred response probabilities that the alignment methods do to the policy model. While "heavily" optimizing the model to align with human preference, the training process overfits the model and weakens its generalization ability. As a side note, the downstream tasks of the models trained by Cal-DPO and RLHF did not show significantly inferior performance compared to SFT. This proves the advantages of DRO.

## 5 RELATED WORK

Large language models (LLMs) have shown great zero-shot and few-shot performance Brown et al. (2020); Chowdhery et al. (2023); Radford et al. (2019). Despite the success of instruction tuning, preference optimization has shown great effectiveness in aligning LLMs with human preferences Bai et al. (2022). As reinforcement Learning with Human Feedback (RLHF) Bai et al. (2022) is a complex and often unstable procedure Pal et al. (2024), DPO Rafailov et al. (2024) has been proposed as a simple and computationally lightweight method with no need for additional reward function training. Specifically, it derives the optimal policy of the RLHF objective and reparameterizes the reward model using the current policy (i.e., using policy as an implicit reward model). In this way, the optimization of the policy model becomes the optimization of the reparameterized reward function using the BT model. This reparameterization method also inspires the variant of RLHF (e.g., K1.5Team et al. (2025)).

Various further methods have been proposed to improve DPO. ORPO Hong et al. (2024) and SimPO Meng et al. (2024) focus on regularization of sequence length to tackle the issue that DPO tends to increase the response length of the policy model. For solving the likelihood displacement problem (discussed in Sec. 1), DPOP Pal et al. (2024), KTO Ethayarajh et al. (2024), and DPO-ShiftYang et al. (2025) aim to lower the preferred response probabilities by increasing the weight of the preferred term in the training objective. However, these methods break the theoretical basis of DPO and obtain uncertain gains. Cal-DPO Xiao et al. (2024) introduces an IPO-like calibration loss as an addition to the DPO loss. Unintentional UnalignmentRazin et al. (2024) provides a data-level approach that contains requirements for data attributes, which adopts the original DPO as the optimization algorithm. In particular, Robust Preference Optimization Fisch et al. (2024) and Reward-Aware Preference Optimization Adler et al. (2024) introduce an explicit general reward model to provide a target reward difference for each prompt. However, they still adopt the pairwise optimization method, which cannot prevent the decreasing chosen reward problem and overlooks the relationship among samples given by the explicit reward model.

Our DRO proposes a point-wise direct alignment method that better utilizes the reward model information and brings a strengthened control over the optimization objective.

## 6 CONCLUSION

In this paper, we propose a Direct Reward Optimization (DRO) method that utilizes a point-wise target for aligning the model.

Compared to the existing direct alignment approaches that are based on pair-wise losses to optimize the policy model. DRO prevents the policy model from dropping the generation probability and utilises a direct target to lead the optimization. Experimental results on various reasoning tasks and datasets demonstrate the superior performance of our DRO which consistently outperforms a wide range of baseline approaches.

## 7 ETHICS STATEMENT

We adhere to the ICLR Code of Ethics. All authors have read and acknowledged the Code during submission. Our study uses publicly available datasets and does not involve human subjects. This work does not involve societal impacts. The authors take full responsibility for the entire content of this paper. There are no potential conflicts of interest to declare.

## 8 REPRODUCIBILITY STATEMENT

The source code is included in our anonymous supplementary materials. Our experiments are based on the publicly available [Specify datasets, e.g., UltraChat, UltraFeedback] datasets. Comprehensive descriptions of our experimental setup, including hyperparameters (such as learning rates, DPO beta values, and model architectures), are detailed in the Appendix and main context of this paper.

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

## A    DERIVING THE OPTIMAL SOLUTION OF RLHF

### A.1    PROOF FOR OPTIMAL SOLUTION OF RLHF

We construct our proof following the previous worksPeters & Schaal (2007); Rafailov et al. (2024). From Eq. 3, our optimizing target is:

$$\max_{\pi} \mathbb{E}_{x \sim \mathcal{D}, y \sim \pi}[r(x,y)] - \beta \mathbb{D}_{\mathrm{KL}}\left[\pi(y \mid x) \| \pi_{\mathrm{ref}}(y \mid x)\right] \tag{9}$$

Notably, we can derive that:

$$\begin{aligned}
\max_{\pi} \mathbb{E}_{x \sim \mathcal{D}, y \sim \pi}&[r(x,y)] - \beta \mathbb{D}_{\mathrm{KL}}\left[\pi(y \mid x) \| \pi_{\mathrm{ref}}(y \mid x)\right] \\
&= \max_{\pi} \mathbb{E}_{x \sim \mathcal{D}} \mathbb{E}_{y \sim \pi(y|x)}\left[r(x,y) - \beta \log \frac{\pi(y \mid x)}{\pi_{\mathrm{ref}}(y \mid x)}\right] \\
&= \min_{\pi} \mathbb{E}_{x \sim \mathcal{D}} \mathbb{E}_{y \sim \pi(y|x)}\left[\log \frac{\pi(y \mid x)}{\pi_{\mathrm{ref}}(y \mid x)} - \frac{1}{\beta} r(x,y)\right] \\
&= \min_{\pi} \mathbb{E}_{x \sim \mathcal{D}} \mathbb{E}_{y \sim \pi(y|x)}\left[\log \frac{\pi(y \mid x)}{\frac{1}{Z(x)}\pi_{\mathrm{ref}}(y \mid x)\exp\left(\frac{1}{\beta}r(x,y)\right)} - \log Z(x)\right]
\end{aligned} \tag{10}$$

where we define as :

$$Z(x) = \sum_{y} \pi_{\mathrm{ref}}(y \mid x)\exp\left(\frac{1}{\beta}r(x,y)\right) \tag{11}$$

Notably, $Z(x)$ is a function of only $x$ and $\pi_{ref}$. We can additionally define:

$$\hat{\pi}(y \mid x) = \frac{1}{Z(x)}\pi_{\mathrm{ref}}(y \mid x)\exp\left(\frac{1}{\beta}r(x,y)\right) \tag{12}$$

As is a probability distribution which holds $\sum_{y} \pi^*(y \mid x) = 1$. Using the $Z(x)$, we can re-organize the Eq. 9 as:

$$\begin{aligned}
\min_{\pi} \mathbb{E}_{x \sim \mathcal{D}} &\left[\mathbb{E}_{y \sim \pi(y|x)}\left[\log \frac{\pi(y \mid x)}{\hat{\pi}(y \mid x)}\right] - \log Z(x)\right] = \\
&\min_{\pi} \mathbb{E}_{x \sim \mathcal{D}}\left[\mathbb{D}_{\mathrm{KL}}\left(\pi(y \mid x) \| \hat{\pi}(y \mid x)\right) - \log Z(x)\right]
\end{aligned} \tag{13}$$

Since $Z(x)$does not depend on $\pi$, the optimal solution is achieved by the policy that minimizes the first term. The KL divergence is minimized in situations where two distributions are equal. Thus, we have the optimal solution:

$$\pi(y \mid x) = \hat{\pi}(y \mid x) = \frac{1}{Z(x)}\pi_{\mathrm{ref}}(y \mid x)\exp\left(\frac{1}{\beta}r(x,y)\right) \tag{14}$$

## B    IMPLEMENT DETAILS

The experiments are carried out on 16 A100-80G GPUs with a Linux system. For all baselines and DRO, we search the hyperparameters as we present the details in the Appendix F. For the SFT phase, we train two epochs in each setting and report the performance of the best checkpoint. We train for three epochs for the alignment phase and take the same approach. We use *Pytorch*[1] and

---

[1]https://pytorch.org/

*Huggingface*[2] as tools for the implementation. For alignment, we apply experiments based on *trl*[3]. All the generations during the evaluation process were done using *vllm* Kwon et al. (2023)[4]. The code will be released on GitHub[5].

## C  DERIVING THE EQUATION 6

We here present our derivation process for Eq. 6. Let's start from Eq. 5. Assuming a reward process on the concatenated response $(y_1, y_2)$ to the prompt $x$ (i.e., the response of $x$ is: $y_1$ followed by $y_2$), the reward of the implicit reward function is:

$$r(x, (y_1, y_2)) = \beta \log \frac{\hat{\pi}\left((y_1, y_2) \mid x\right)}{\pi_{\text{ref}}\left((y_1, y_2) \mid x\right)} + \beta \log Z(x) \tag{15}$$

Then, we can observe this equation from another perspective. Rewarding the prompt of $x, y_1, y_2$ as $(x, y_1)$ and the response is $y_2$, the the reward of the implicit reward function is:

$$r((x, y_1), y_2)) = \beta \log \frac{\hat{\pi}\left(y_2 \mid (x, y_1)\right)}{\pi_{\text{ref}}\left(y_2 \mid (x, y_1)\right)} + \beta \log Z(x, y_1) \tag{16}$$

since $r(x, (y_1, y_2)) = r((x, y_1), y_2))$, we can derive from combining Eq. 15 and Eq. 16 into:

$$\beta \log \frac{\hat{\pi}\left(y_2 \mid (x, y_1)\right)}{\pi_{\text{ref}}\left(y_2 \mid (x, y_1)\right)} + \beta \log Z(x, y_1) = \beta \log \frac{\hat{\pi}\left((y_1, y_2) \mid x\right)}{\pi_{\text{ref}}\left((y_1, y_2) \mid x\right)} + \beta \log Z(x)$$

$$\log \frac{\hat{\pi}\left(y_2 \mid (x, y_1)\right)}{\pi_{\text{ref}}\left(y_2 \mid (x, y_1)\right)} - \log \frac{\hat{\pi}\left((y_1, y_2) \mid x\right)}{\pi_{\text{ref}}\left((y_1, y_2) \mid x\right)} = \beta \log Z(x) - \log Z(x, y_1) \tag{17}$$

$$\frac{Z(x, y_1)}{Z(x)} = \frac{\hat{\pi}(y_1 \mid x)}{\pi_{ref}(y_1 \mid x)}$$

Thus, we derive Eq. 6.

## D  PROOF OF THM. 3.1

Suppose a reward model $r(x, y)$ and a response dataset $\mathcal{D} = \{x_i, y_i\}_{i=1}^{N}$. We can construct a reward function as:

$$\bar{r}_{r, \alpha_1, \alpha_2, \mathcal{D}}(x_i, y_i) = \begin{cases} \alpha_1 \left[r(x_i, y_i) - \delta_{\mathcal{D}}^{max}\right] + \delta_{\mathcal{D}}^{max} & \text{if } r(x_i, y_i) > \delta_{\mathcal{D}}^{max} \\ \delta_{\mathcal{D}}^{min} - \alpha_2 \left[\delta_{\mathcal{D}}^{min} - r(x_i, y_i)\right] & \text{if } r(x_i, y_i) < \delta_{\mathcal{D}}^{min} \\ r(x_i, y_i) & \text{if } \delta_{\mathcal{D}}^{min} < r(x_i, y_i) < \delta_{\mathcal{D}}^{max} \end{cases} \tag{18}$$

where

$$\delta_{\mathcal{D}}^{max} = max\left(\{r(x_i, y_i) | (x_i, y_i) \in \mathcal{D}\}\right)$$
$$\delta_{\mathcal{D}}^{min} = min\left(\{r(x_i, y_i) | (x_i, y_i) \in \mathcal{D}\}\right) \tag{19}$$

With different setting of $\alpha_1$ and $\alpha_1$ ensuring $\alpha_1, \alpha_2 > 0$, different reward model $\bar{r}(r, \alpha_1, \alpha_2, \mathcal{D})$ can be constructed. Each constructed reward function $\bar{r}$ satisfies that the reward sorting of $\bar{r}$ is the same with $r$ and the reward value of the responses in $\mathcal{D}$ is invariable (i.e., $\bar{r}_{r, \alpha_1, \alpha_2, \mathcal{D}}(x_i, y_i) = r(x_i, y_i)$). This proves the Theorem. 3.1. Specifically, different $\alpha_1$ and $\alpha_1$ lead to different $Z_0$, thus regarding $Z_0$ as a hyperparameter doesn't detract from the theoretical basis.

---

[2]https://huggingface.co/

[3]https://github.com/huggingface/trl

[4]https://github.com/vllm-project/vllm

[5]http://github.com/xxxxxx

# E DATASETS

•UltraChat-200k is a multi-turn instructional conversation dataset that contains 207,865 conversations for training. UltraChat-200k is designed by a principle that attempts to capture the breadth of interactions that a human might have with an AI assistant and then employs meta-information, in-context expansion, and iterative prompting to scale up the number of instructions. The constructors use LLMs only to generate the conversations.

•UltraFeedback is a large-scale, high-quality, and diversified AI feedback dataset, which contains over 1 million GPT-4 feedback for user-assistant conversations from various aspects. It is constructed beyond a compiled diverse array of over 60,000 instructions and 17 models from multiple sources and then utilizes GPT-4 for annotation with a bunch of techniques to alleviate annotation biases and improve feedback quality to the greatest extent. Notably, we utilize binary preferences from UltraFeedback by selecting the highest mean score as the preferred response and one of the remaining three at random as dispreferred, referring to Tunstall et al. (2023). The total number of data pairs for training is 61,135.

# F HYPERPARAMETER SEARCH

Table 4: Hyperparameter search range.

| Methods | Search Range |
|---------|--------------|
| **DPO** | $\beta \in [0.05, 0.1, 0.5, 1.0]$ |
|  | $lr \in [1e-7, 2e-7, 5e-7, 1e-6]$ |
| **SLiC-HF** | $\lambda \in [0.05, 0.1, 0.5, 1.0, 5, 0]$ |
|  | $lr \in [1e-7, 2e-7, 5e-7]$ |
| **IPO** | $\beta \in [0.05, 0.1, 0.5, 1.0]$ |
|  | $lr \in [1e-7, 2e-7, 5e-7, 1e-6]$ |
|  | $\alpha \in [0.25, 0.5, 1, 2]$ |
| **ORPO** | $\tau \in [0.01, 0.05, 0.1, 1.0]$ |
| **SimPO** | $\beta \in [1.0, 2.0, 2.5]$ |
|  | $\gamma \in [0.3, 0.5, 0.7, 1.0, 1.5]$ |
| **RPO** | $\beta \in [0.05, 0.1, 0.5, 1.0]$ |
| **DRO** | $\beta \in [0.05, 0.1, 0.5, 1.0]$ |
|  | $lr \in [1e-7, 2e-7, 5e-7, 1e-6]$ |
|  | $Z_0 \in [-50, 500]$ |

Notably, we are referring to the papers Rafailov et al. (2024); Meng et al. (2024); Hong et al. (2024); Azar et al. (2024); Zhao et al. (2023) to set the search ranges. For our DRO, we report the results under learning rate 5e-7, beta 0.1 with $Z_0$ 50 in our paper (which is also mentioned in the Experiments Section in the paper).

# G LLM USAGE

We disclose that large language models (GPT) were used for grammar checking and polishing text in this manuscript (i.e., Grammaly). There's no further usage of LLM in this paper.

