# OpenReview forum: "Direct Reward Optimization: A Point-wise Alignment Approach"
_ICLR.cc/2026/Conference — ICLR 2026 Conference Withdrawn Submission_

### Official Review · Reviewer_seTC · 2025-10-30

**Soundness:** 2
**Presentation:** 3
**Contribution:** 1
**Rating:** 2
**Confidence:** 4

**Summary:**

This paper proposes Direct Reward Optimization (DRO), which adopts an explicit reward function to distill it into the language model (LM) policy. It introduces a novel estimation of the normalization factor $Z(x)$ to address the likelihood displacement problem and better leverage relationships among responses across different prompts. Empirical results demonstrate that DRO outperforms existing methods.

**Strengths:**

- The proposed estimation of normalization factor is novel and interesting.

**Weaknesses:**

* **Consistency between DPO and RLHF:** The consistency between the DPO and RLHF objectives (i.e., both objectives converging to the same closed-form optimal policy) rarely holds for standard DPO in practice, since it requires a strong assumption that the offline dataset fully covers the entire prompt–response space (\mathcal{X} \times \mathcal{Y}) to achieve the closed-form solution [1,2]. It remains unclear how DRO resolves this consistency issue between DPO and RLHF, given a finite offline dataset, compared to prior approaches. Furthermore, the additional weighting mechanisms in prior works are well-justified under the *principle of pessimism* [2], encouraging the LM policy to remain close to the offline distribution. In addition, DRO uses a global prefix normalization factor to better utilize information from different prompts and responses. However, this global normalization factor is extremely difficult to estimate and model-dependent. The paper simply treats it as a hyperparameter with an arbitrary range and provides no principled method to estimate it or analyze how it balances the bias–variance trade-off in estimating the per-prompt normalization factor ($Z(x)$).

* **Contrast to other explicit-reward approaches:** The paper claims that prior explicit-reward approaches do not account for the drop in the probabilities of preferred responses. However, Fisch et al. (2024) employ pessimistic reward distillation via an explicit forward KL regularization toward the reference model, which effectively mitigates likelihood displacement and reward over-optimization in DAAs. It is unclear how DRO provides a more principled or effective solution to this issue compared to prior explicit-reward methods.

* **On the Bradley–Terry assumption for learning complex reward functions:** The paper argues that the Bradley–Terry assumption is inadequate for learning complex tasks (e.g., mathematical reasoning) and cannot make use of chain-of-thought reasoning. While it is true that DAAs cannot explicitly utilize CoT traces, DAAs can still leverage fine-grained scalar information from a reward model. More specifically, we can define a soft-label objective with preference probabilities:
 $$
\mathcal L_{\text{DPO-RM}}(\pi_\theta) = -\mathbb E_{(x,y_w,y_l)\sim\mathcal D}\left[w\log\sigma(r^{\text{implicit}}(x,y_w)-r^{\text{implicit}}(x,y_l)) + (1-w)\log\sigma\right]
$$

Thus, DPO can still utilize detailed scalar feedback from an explicit reward model, similar to DRO. Moreover, the paper does not provide sufficient details on how DRO can meaningfully leverage additional CoT information in scenarios where DPO cannot.
* **Lack of novelty:** Likelihood displacement and explicit-reward methods have been extensively studied and well-explained [3,4]. Pointwise approaches have also been widely explored (see [5,6]). In particular, [6] proposes a pointwise implicit-reward method with an MSE loss that closely resembles DRO, with the primary difference being the estimation of the normalization factor, which appears too incremental to me.
## References
[1] Scaling Laws for Reward Model Overoptimization in Direct Alignment Algorithms. NeurIPS 2024.

[2] Contrastive Preference Learning: Learning from Human Feedback without Reinforcement Learning. ICLR 2024.

[3] Learning Dynamics of LLM Finetuning. ICLR 2025 Oral.

[4] Unintentional Unalignment: Likelihood Displacement in Direct Preference Optimization. ICLR 2025.

[5] REBEL: Reinforcement Learning via Regressing Relative Rewards. NeurIPS 2024.

[6] Offline Regularised Reinforcement Learning for Large Language Models Alignment. CoRR 2024

**Questions:**

See weaknesses

---

### Official Review · Reviewer_2UWt · 2025-10-31

**Soundness:** 2
**Presentation:** 2
**Contribution:** 2
**Rating:** 2
**Confidence:** 5

**Summary:**

The paper proposes Direct Reward Optimization (DRO), a new approach for direct alignment that addresses several known limitations of existing Direct Alignment Algorithms (DAA), such as the likelihood shift caused by pairwise optimization. DRO introduces a KL-regularized objective that directly maximizes the expected reward, aiming to decouple reward differential from bias and thereby stabilize the learning of human preference models. The method is designed to prevent the undesirable reduction of preferred response probabilities that can occur in pairwise formulations.

**Strengths:**

The approach is simple enough to integrate with existing alignment pipelines, making it a potentially useful alternative to DPO-like objectives.

**Weaknesses:**

1.  In Equation (5), the baseline term b is not well specified. How is it estimated—empirical mean, moving average, or a learned baseline? The training stability might heavily depend on this choice.
2. Treating the normalization constant as a tunable hyperparameter seems ad hoc, given that normalization in policy gradients primarily serves variance reduction rather than optimization tuning.
3.  Since the paper emphasizes normalization as a key design choice, it should include systematic experiments showing how different normalization strategies affect stability and final performance.

**Questions:**

see the weakness.

---

### Official Review · Reviewer_kxQq · 2025-10-31

**Soundness:** 2
**Presentation:** 2
**Contribution:** 2
**Rating:** 2
**Confidence:** 4

**Summary:**

The paper proposes Direct Reward Optimization (DRO), a point-wise alignment algorithm for large language models that aims to overcome the limitations of pairwise methods such as DPO. Instead of optimizing reward differences between preferred and dispreferred responses, DRO directly aligns the model’s output probabilities with explicit reward scores provided by a reward model. The approach treats the normalization term in the RLHF objective as a global hyperparameter $Z_0$, enabling a simple MSE-based offline optimization. Experiments on Llama-3, Qwen-2.5, and EuroLLM show that DRO improves performance on AlpacaEval-2, MT-Bench, and several reasoning benchmarks while avoiding degradation of reasoning ability.

**Strengths:**

- The paper presents a simplified formulation of RLHF-style objectives by analytically decomposing the normalization term and showing that it can be treated as a single scalar $Z_0$.
- The resulting method is relatively easy to implement (based on an explicit reward model and a point-wise MSE objective).
- Experiments on several base models (Llama-3, Qwen-2.5, EuroLLM) demonstrate consistent improvements over other direct alignment methods across multiple benchmarks.

**Weaknesses:**

-  The paper does not cite Richemond et al. ("Offline Regularised Reinforcement Learning for LLM Alignment" 2024), which already proposed a method also named Direct Reward Optimization (DRO) based on a similar derivation from the RLHF objective. The main difference seems to lie in how the normalization term is treated (Richemond estimates it, while this paper decomposes it analytically and replaces it with a hyperparameter $Z_0$), but it remains unclear whether this difference provides any clear advantage in alignment performance or theoretical soundness. A detailed comparison highlighting the novelty of the present work would be crucial.
- Although Theorem 3.1 appears to support treating $Z_0$ as a hyperparameter, its implication for the proposed algorithm is unclear. The theorem mainly establishes the existence of equivalent reward models, but does not clarify what properties of convergence or optimality it guarantees for the proposed method DRO.
- The theoretical motivation relative to RLHF is incomplete. While the paper clearly contrasts DRO with DPO and its variants, it lacks a discussion of how DRO theoretically compares to or improves upon RLHF, given that both methods use the same explicit reward model and optimize a similar objective. This omission weakens the paper's theoretical soundness.
- The procedure for selecting hyperparameters (especially $Z_0$ ) is unclear. While Appendix F lists the search ranges, the criteria for choosing the final values are not explained, e.g., which dataset or evaluation metric (AlpacaEval, MT-Bench, etc.) was used for selection.
- The paper does not report error bars, confidence intervals, or
results across multiple random seeds. It appears to be unclear whether
the observed differences are statistically significant or simply
due to random variation.

**Questions:**

Questions

- What exactly does Theorem 3.1 guarantee for the proposed method?
- The proposed method consistently outperforms RLHF, even though both rely on an explicit reward model. Could the authors provide a mathematical explanation/discussion for why DRO would surpass RLHF in this setting?
- How were the hyperparameters determined? For example, which dataset and evaluation metric were used for tuning?


Additional comments/suggestions

- The method name Direct Reward Optimization (DRO) was already used by Richemond et al. (2024). To avoid confusion, the authors may consider adopting a different name.
- Richemond et al. (2024) approximated the normalization term as a learnable function of the prompt x, while the present paper decomposes it analytically and replaces it with a scalar hyperparameter Z_0. Instead of fixing Z_0, would it be feasible to treat it as a learnable parameter jointly optimized with the policy, similar in spirit to Richemond et al. (2024)? This could preserve simplicity while avoiding manual tuning.
- GRPO (Group Relative Policy Optimization) has become a standard baseline in LLM alignment; including it would strengthen the empirical evaluation.
- Extending the analysis in Figure 2(d) to other model and dataset settings would provide more practical implications for how the hyperparameter Z_0 influences model performance and alignment behavior.
- Line 322: "with the base model the SFT model and the implicit reward model" should be "with the base model and the SFT model, and the implicit reward model" or similar.

Reference
- Richemond et al.,  Offline Regularised Reinforcement Learning for Large Language Models Alignment, 2024.

---

### Official Review · Reviewer_nfzN · 2025-11-01

**Soundness:** 2
**Presentation:** 2
**Contribution:** 2
**Rating:** 4
**Confidence:** 3

**Summary:**

This paper introduces Direct Reward Optimization (DRO), a alignment method for LLMs. Unlike existing Direct Preference Optimization (DPO) approaches that rely on pairwise Bradley–Terry (BT) training, DRO employs an explicit reward model and a point-wise (per-sample) optimization objective.
The key idea is to directly guide the generation probability of each response using a target derived from the explicit reward, thereby preventing the issue in DPO where the probabilities of preferred responses unintentionally decrease.
A normalization constant Z_0 is treated as a tunable hyperparameter, enabling control over the offset of the reward distribution.

**Strengths:**

1. The paper provides clear derivations from the RLHF objective, linking explicit reward distillation to implicit policy optimization.

2. Evaluations cover multiple baselines (DPO, RLHF, RPO, SimPO, ORPO, etc.) and datasets.

**Weaknesses:**

1. heuristically derived quantity (e.g., via the Z(x,y)/Z(x)=\hat\pi(y|x)/ \pi_{\text{ref}}(y|x)  relation and the Z_0 offset). In practice this can yield inaccurate probability targets and introduce systematic bias/miscalibration, especially when reward distributions vary across prompts or are off-distribution. Unlike pairwise BT/DPO losses, where Z(x) cancels, DRO’s misspecified Z directly distorts the absolute targets, so its “point-wise is better than pairwise” claim may not hold under normalizer misspecification.

2. The role of Z_0 and the assumption of a shared prefix t_0 are mathematically justified but lack clear intuitive motivation.The derivations may be difficult to follow for readers without strong theoretical background.

3. While DRO is empirically compared against Cal-DPO and other DPO variants in the experiments, the theoretical differentiation remains underexplored. Both methods introduce mechanisms to calibrate or rescale reward estimates and normalization terms, yet the paper does not rigorously analyze where DRO’s formulation departs fundamentally from these existing approaches.

**Questions:**

The role of Z_0 and the assumption of a shared prefix t_0 are mathematically justified, but their intuitive motivation remains unclear. I did not fully understand the mechanism by which Z_0 approximates or replaces the true normalization term Z(x), nor why assuming a shared prefix t_0 across prompts is theoretically reasonable. Could the authors elaborate on how Z_0 is computed or estimated in practice, and provide more intuition for why this approximation holds empirically?

---

### Note · Authors · 2025-11-13

**Comment:**

After careful discussion, we decide to withdraw this paper

**Withdrawal Confirmation:**

I have read and agree with the venue's withdrawal policy on behalf of myself and my co-authors.